# Applications of Polyacetylene Derivatives in Gas and Liquid Separation

**DOI:** 10.3390/molecules28062748

**Published:** 2023-03-18

**Authors:** Manyu Chen, Guangze Hu, Tanxiao Shen, Haoke Zhang, Jing Zhi Sun, Ben Zhong Tang

**Affiliations:** 1MOE Key Laboratory of Macromolecular Synthesis and Functionalization, Department of Polymer Science and Engineering, Zhejiang University, Hangzhou 310027, China; 2Centre of Healthcare Materials, Shaoxing Institute, Zhejiang University, Shaoxing 312000, China; 3Hangzhou Global Scientific and Technological Innovation Center, Zhejiang University, Hangzhou 311215, China; 4Shenzhen Institute of Aggregate Science and Technology, School of Science and Engineering, The Chinese University of Hong Kong, Shenzhen 518172, China

**Keywords:** membrane separation, substituted polyacetylenes, gas permeability, aging

## Abstract

As a low energy consumption, simple operation and environmentally friendly separation method, membrane separation has attracted extensive attention. Therefore, researchers have designed and synthesized various types of separation membrane, such as metal organic framework (MOF), covalent organic framework (COF), polymer of intrinsic micro-porosity (PIM) and mixed matrix membranes. Some substituted polyacetylenes have distorted structures and formed micropores due to the existence of rigid main chains and substituted side groups, which can be applied to the field of membrane separation. This article mainly introduces the development and application of substituted polyacetylenes in gas separation and liquid separation based on membrane technology.

## 1. Introduction

With the improvement in people’s living standards and the growth of the population, people’s demand for energy is growing, which greatly promotes the rapid development of the chemical industry. However, the chemical industry inevitably releases carbon monoxide, carbon dioxide, volatile organic compound (VOC) gas and a large number of organic solvent mixture while achieving high value output, leading to global warming, water pollution and severe weather with frequent disasters. Recently, the UN Paris Agreement on Global Warming came into force, it is required that global greenhouse gas emissions should be reduced to 32 billion tons by 2030 [1]. However, the development of the chemical industry and other industries cannot be completely prohibited, which requires the development of gas separation and capture technology. To build an ecologically civilized society, it is necessary to find ways to separate mixed gases and organic liquids with low energy consumption, low cost and high efficiency.

In the field of gas separation, traditional separation methods include solvent absorption, low temperature distillation and pressure swing adsorption. Although the solvent absorption method can obtain high separation efficiency by selecting the corresponding solvent for different gases, it is inevitable that the solvent needs post-treatment and the energy consumption is large. Additionally, the separation methods of low temperature distillation and pressure swing adsorption have high energy consumption and cost. However, membrane separation technology can achieve high permeability and high separation by rationally selecting different substances for different gases, designing molecular structures and surface modifications. It has the advantages of high separation efficiency, low energy consumption, a small footprint, simple operation and no phase change [2]. In the chemical and pharmaceutical industries, organic solvents are inevitably used. Nevertheless, organic solvents cannot be directly discharged because of their toxicity and damage to the environment. Traditional separation and treatment methods such as distillation, adsorption and extraction have high energy consumption and huge costs. Therefore, membrane separation technology has become an excellent method in the field of gas separation and organic solvent separation due to its low energy consumption, low cost and simple operation.

In recent years, membrane separation technology has flourished. Researchers are moving towards the goal of high selectivity, low energy consumption and low cost. Based on the relationship between structure and performance, different types of separation membranes such as organic metal framework (MOF), covalent organic framework (COF) and polymer of intrinsic microporosity (PIM) were obtained by designing a molecular structure [3,4,5,6,7]. Since PTMSP was found to have ultra-high gas permeability, substituted polyacetylenes have become a promising candidate as permeable membranes for gas and liquid separation. Some substituted polyacetylenes show superior performance in gas permeability due to both their stiff main chain composed of alternating double bonds and the steric repulsion of spherical side groups. However, there are many restrictions on the development of gas separation membranes. Thus, significant efforts have been devoted to improve selectivity performance of substituted polyacetylenes. This article will mainly introduce the development of substituted polyacetylenes in gas and liquid separation. Some existing studies on improving selectivity will be discussed, as well as aging mechanisms and improvement method, and the prospect of future development will be put forward.

## 2. Separation Mechanism

The separation of gas by membrane separation technology is driven by the difference in pressure or concentration on both sides. Utilizing the different speed of different gases passing through the membrane, the gas with a fast permeation rate is enriched on the permeation side, and the gas with a slow permeation rate is enriched on the feed side, thus achieving gas separation.

Gas separation membranes can be roughly divided into two types: porous membranes and dense membranes. The gas transfer process can be explained by the pore transfer model [8,9] and the solution–diffusion mechanism, respectively [10,11,12]. For pore transport [12], it can be understood as the molecular thermal motion of gas molecules in membrane pores driven by pressure. According to the pore size (*dp*) and the average free path (*λ*) of molecular motion, it can be divided into Knudsen flow and viscous flow. When the collision probability between the gas molecules and the pore wall is much larger than the collision probability between the gas molecules, the gas is dominated by the Knudsen motion. The gas flux is inversely proportional to the molecular weight, so the separation effect can be achieved due to the different molecular weight of the gas molecules. At the same time, it has to be considered that there is a certain interaction between different gases and membrane materials, that is, adsorption and diffusion. Gas separation can also be achieved based on the difference in adsorption performance. According to the separation mechanism of pore transfer, it is not difficult to see that the effect of pore transfer to separate gas is limited, due to the difficulty of separating molecules with a similar molecular weight and adsorption properties. In the dense membrane, the gas transfer is driven by the concentration difference, and the mass transfer and separation are carried out according to the dissolution–diffusion mechanism. The gas dissolves on the surface of the feed side, and then diffuses from high concentration to low concentration in the membrane under the driving force of concentration difference, and desorbs from the membrane after reaching the other side. The process of dissolution and diffusion follows Henry’s law and Fick’s law, respectively. Moreover, the product of the dissolution coefficient (*S*) and the diffusion coefficient (*D*) is defined as the permeability coefficient (*P*), which represents the ability of the gas to pass through the membrane. The ratio of the permeability coefficient of the two gas molecules is defined as the separation coefficient (*α*) of the membrane to the two gases [13,14,15]. The permeability coefficient and the separation coefficient characterize the gas separation ability of the membrane [12,16]. The gas permeability (*P*) related to gas diffusivity (*D*) and gas solubility (*S*) is demonstrated in Equation (1). Moreover, the separation coefficient is determined by dividing the permeability of gas *i* to the permeability of gas *j* as shown in Equation (2).
(1)P=D×S
(2)$$\alpha_{i/j}=\frac{P_{i}}{P_{j}}$$

Furthermore, separation membranes for liquids can be classified into microfiltration membrane (0.1~10 μm), ultrafiltration membrane (1 nm~0.2 μm), nanofiltration membrane (1~5 nm) and reverse osmosis membrane (<1 nm) according to the pore size, and correspond to different separation mechanisms. As with the separation membrane, solvent separation and gas separation membranes have a similar working mechanism. The separation of microfiltration and ultrafiltration membrane is mainly based on the pore-size screening mechanism. As for the separation mechanism of nanofiltration and reverse osmosis, there are various descriptions, including irreversible thermodynamics, pore flow model and solution diffusion model [17,18]. The solution diffusion theory proposed by Lonsdale et al. in 1965 [19] when studying cellulose acetate membranes was widely used. This theory holds that the solvent and solute are independent and can be dissolved in a dense membrane, and then diffuse through the membrane under concentration or pressure driving force. Based on the efficient desalination properties of nanofiltration and reverse osmosis membranes, nanofiltration and reverse osmosis membranes are currently mainly used in seawater desalination, ultrapure water preparation, wastewater treatment and other fields. Pervaporation is also an effective method for separating liquid mixtures. Based on the dissolution–diffusion mechanism, each component is selectively adsorbed and dissolved by the membrane under the promotion of vapor pressure on both sides of the membrane, diffuses at different speeds in the membrane, and vaporizes and desorbs at the permeation side of the membrane, thereby achieving separation of the mixture. 

Different from nanofiltration and reverse osmosis based on concentration or pressure as the driving force, pervaporation uses vapor pressure difference as the driving force. The vapor pressure difference on both sides of the membrane can be tested by vacuum pervaporation, hot pervaporation and carrier gas-purge pervaporation. At present, pervaporation is mainly used for azeotrope separation, organic solvent dehydration, removal of trace organic matter in water and separation of organic mixtures. Whether gas separation membrane or liquid separation membrane, the most important is to have both high permeability and high selectivity. However, just as the proverb says, “you can’t have your cake and eat it”, the upper bound relationship between permeability and selectivity proposed by Robeson in 1991 [20] and supplemented in 2008 [21] reveals a “reciprocal” relationship between the two (see Figure 1). Therefore, the preparation of separation membranes with high permeability and selectivity is a great challenge for researchers.

## 3. Application of Substituted Polyacetylenes in Gas Separation 

Polymers are currently the most studied and widely used membrane materials. From the earliest cellulose acetate, polysulfone, polydimethylsiloxane, polyimide, polyester, polyacetylene and other gas separation membrane materials have been developed. These polymers can be divided into rubbery and glassy polymers. Polydimethylsiloxane (PDMS) and polyethylene glycol (PEG) are typical rubbery molecules with extremely high solubility coefficient of carbon dioxide. PDMS is the polymer with the best permeability for a long time. Until the emergence of poly (1-trimethylsilyl-1-propyne) (PTMSP), the oxygen permeability coefficient of PTMSP is as high as 6000 barrer, which is ten times that of PDMS. Unlike traditional glassy polymers, PTMSP has higher permeability to organic solvents and gases, which may be due to the rigid main chain and large substituents leading to the formation of micropores in PTMSP with a high free volume [22]. This type of polymer of intrinsic microporosity (PIM) has a distorted stereo conformation, and the micropores smaller than 2 nm mean that the polymer has a larger free volume, thus improving the permeability to gas. By designing the molecular structure to adjust the free volume, the introduction of different groups can change the adsorption of different molecules, which is expected to obtain high permeability and high selectivity of the separation membrane. Next, we will describe the development of substituted polyacetylenes in the application of separation membrane, and put forward a prospect.

Polyacetylene, as the prototype of conductive polymers, has caused a great sensation, which can be synthesized through the following route (see Figure 2), but it is difficult to put into use due to its insolubility, infusibility and instability. Subsequently, researchers have introduced different substituents to improve its performance, and in the process obtained properties such as liquid crystal, magnetic, photoluminescence, etc. In 1983, Masuda et al. found the high permeability of PTMSP, and the oxygen permeability, was much higher than that of the most permeable polymer (polydimethylsiloxane) at that time, which expanded the application of substituted polyacetylenes in gas separation [23,24,25,26,27].

Following this pioneering discovery, researchers have studied the permeability mechanism of PTMSP. After experimental verification, the curve of gas concentration and osmotic pressure in PTMSP presents a concave shape, that is, the adsorption isotherm of small molecule gas in glassy polymer, which can be explained by double adsorption model [28,29]: (1) The dissolution mechanism of gas in the more relaxed region is similar to that in rubbery polymer, which conforms to Henry’s law. (2) In the non-relaxation glassy region of micropores, the adsorption model following the Langmuir isotherm plays a major role. This also provides ideas for the development of more permeable substituted acetylene. Ichiraku et al. [30] studied and analyzed the reasons for the high permeability of PTMSP. Through experiments, compared with PDMS, PTMSP has a very high solubility for oxygen, carbon dioxide, nitrogen and other gases, which may be attributed to the very high free volume of the polymer. The chemical structure determines that the polymer has a high free volume. The alternating carbon–carbon double bonds make the PTMSP have a relatively high, rigid main chain. The large spherical trimethyl silane substituent limits the movement of the segments, resulting in a high free volume in the non-relaxation region. 

Molecular dynamics’ simulation can help researchers to predict and analyze the experimental results to a certain extent. For example, Catlow et al. [31] obtained potential parameters by using bond energy, vibration data and structural data to study the original form of trans-polyacetylene and its mobility under different doping. Molecular dynamics simulation can also be used to analyze the gas separation performance of substituted acetylene. Clough et al. [32] calculated the single-bond rotational potential barrier of several substituted acetylenes by molecular dynamics’ simulation and calculated that the chain conformation became non-planar with the increase in the volume and number of side groups. According to their results, the rotation potential barrier of TMSP around the single bond of PTMSP main chain is about 40 kcal/mol. Thus, the torsional chain conformation will be maintained at room temperature. The dimension of the chain will change significantly only through the torsional wave in the narrow energy well. In a word, molecular dynamics’ simulation is an effective means to design a separation-capable substitute for polyacetylenes.

In addition to the extended research on PTMSP, researchers have also explored other disubstituted polyacetylenes with permeability based on the strategy of large substituents and rigid main chains. By introducing spherical substituents, such as trimethylsilyl, trimethyl germanyl and tert-butyl, similar structures and distorted conformations can be obtained. Large benzene ring substituents can hinder chain stacking and make the resulting polymers have a high free volume, such as poly[1-(trimethyl germanyl)-1-propyne] (PTMGP), poly(4-methyl-2-pentyne) (PMP), poly[1-phenyl-2-(p-trimethylsilyl) ethynylene] (PTMSDPA)(see Figure 3) and other permeable disubstituted acetylenes which were summarized by Masuda et al. [22,33,34,35]. In summary, the main goal of improving the gas separation performance of polymers is to increase selectivity based on high gas permeability [20] and to solve the problem of the aging of glassy polymers with high free volume (e.g., PTMSP [36]).

There are two main methods to improve selectivity. One is to obtain a mixed membrane by mixing multiple components, and the other is to obtain a self-supporting membrane by chemical modification, which includes halogenation, surface cross-linking, copolymerization and other methods. Taking PTMSP as an example, the composite membrane obtained by coating a highly selective polybenzodioxane (PIM-1) on a highly permeable cross-linked PTMSP by Ilya et al. [37] has high CO_2_/N_2_ selectivity. The separation factor *α* = 35.8–55.7, which is higher than PIM-1 (*α* = 18.5) and cross-linked PTMSP (*α* = 3.7). Among them, Viktoriya et al. [38] modified PTMSP by introducing ionic-liquid butyl imidazolium bromide with high CO_2_ solubility as a sub-substituent. The selectivity was doubled and the permeability was not reduced much, which was closer to the upper limit curve in the Robeson diagram than the original PTMSP. On the basis of the previous introduction of bromine and fluorine atoms, Kossov et al. [39] successfully improved the selectivity to CO_2_ by introducing chlorine atoms on the side chains of PTMSP and PMP, but the permeability was reduced compared with the initial polymer. This is consistent with Robeson’s law, that is, there is a “trade-off” relationship between permeability and selectivity.

To tackle the aging problem of glassy/microporous polymers, one first needs to understand the aging mechanism. Taking porous polymer PTMSP as an example [40], its aging mechanism can be divided into three parts: (1) Physical aging, which means with the passage of time, the polymer chains relaxes, free volume decreases and bulk density becomes larger, resulting in a significant reduction in permeability; (2) Absorption aging. The polymer membrane absorbs nonvolatile impurities resulting in a decrease in free volume. (3) Chemical aging. This means that the polymer absorbs impurities in the environment to form oxygen-containing groups. At high temperatures, the main chain even breaks to form a low-molecular-weight oxygen-containing polymer. This polar oxygen-containing group leads to an increase in bulk density and poor polymer permeability. Moreover, it is found that the thinner the film, the faster the aging rate, because more free volume is exposed to the surface, which will relax faster. Physical aging is very common in porous polymers with high free volume. Although physical aging can be eliminated by dissolving and re-forming the membrane, absorption aging can also be eliminated by dissolving, reprecipitating and then dissolving the reprecipitate and re-forming the membrane. However, these processes will undoubtedly increase energy consumption and cost, so aging still provides a major resistance to commercial membranes. 

So, how to avoid aging? According to the aging mechanism, it can be found that inhibiting chain stacking or “freezing the polymer high free volume” is theoretically an effective way to attenuate aging. At present, there are two main ways, one is to improve the rigidity of the polymer structure through the design of the structure or cross-linking, copolymerization and other modification methods; the other is to support microporosity by adding additives [41]. However, rigid chemical structures and cross-linking methods that increase the rigidity of the polymer do not necessarily improve aging, because rigid polymer chains are not equivalent to restricted chain motion [42,43]. Therefore, additives are usually added to prevent pore collapse and maintain high free volume. In 2014, Lau et al. [44] demonstrated that tetrakis(4-bromophenyl) methane self-condensed to form carbon-based microporous arrays (PAF-1) (see Figure 4) through the Yamamoto coupling reaction. PAF-1 is a porous aromatic skeleton particle with a pore diameter of about 1.2 nm, which is attractive for polyacetylene side chains or large-volume chemical structures. On the other hand, the similar attraction of chemical groups makes the aromatic hydrogen energy of PAF-1 and the methyl group of PTMSP attract each other. The fine insertion of PTMSP in PAF-1 can be seen from Figure 4C,D below, thereby freezing the polymer structure in an appropriate position and improving aging. With the increase in time, PAF-1 can prevent the collapse of the pores, but the large pore size becomes smaller, so that large gas molecules cannot pass through, small molecule gas is not affected, which is conducive to H_2_/N_2_ separation [45]. However, the difficulty of synthesizing PAF-1 makes the membrane costly and difficult to apply. In order to reduce costs, Lau et al. [46] developed a hyper-crosslinking additive (p-DCX) similar to the structure of PAF-1 in 2016, reducing costs on the basis of improving aging (see Figure 5).

## 4. Application of Substituted Polyacetylenes in Liquid Separation 

PTMSP not only has high permeability, but also has the ability to separate liquids. In 1986, Ishihara and Nagase et al. found that the hydrophobic PTMSP membrane could effectively separate the mixture of ethanol and water, and the separation factor of ethanol and water was 11.2 [47]. Subsequently, by means of pervaporation, Nagase et al. carried out a series of chemical modifications on PTMSP from 1989 to 1991 to improve its separation performance, such as PTMSP and PDMS graft copolymerization [48,49]. When the PDMS content was 12%, the polymer had good ethanol permeability selectivity, and the separation factor of ethanol and water was as high as 28.3. When the PDMS content is more than 60%, oxygen and nitrogen separation performance remained stable for more than one month. Furthermore, through metallization and trialkylchlorosilane alkylation, a trialkylsilyl group [50] was introduced to PTMSP. It was found that most of the alkylsilylated membranes had higher selectivity for ethanol and water than PTMSP, which may be due to the increase in hydrophobicity of the membrane. In addition, trimethylsilylated PTMSP membrane can effectively separate ethanol, acetone, acetonitrile, dioxane and isopropanol from an aqueous solution under pervaporation conditions. 1-Trimethylsilyl-1-propyne (TMSP) was copolymerized with 1-(3,3,3-trifluoropropyldimethylsilyl)-1-propyne (FPDSP) or 1-(1*H*,1*H*,2*H*,2*H*-trifluorooctyldimethylsilyl)-1-propyne (FODSP) to introduce fluoroalkyl groups. The introduction of about 5 mol% fluoroalkylsilylation units into PTMSP improved the hydrophobicity without destroying the membrane structure of PTMSP and notably improved the selectivity. The TMSP/FPDSP copolymer membrane can effectively separate tetrahydrofuran, acetone, acetonitrile, dioxane and isopropanol dilute aqueous solution [51], and PTMSP and PDMS composite membrane [52]. 

The nanoporous hydrophobic polymer structure means that PTMSP can be applied to organic solvent nanofiltration. Volkov et al. [53,54,55] proved the feasibility of PTMSP as a solvent-resistant nanofiltration. A series of studies on liquid–liquid separation of PTMSP was carried out, and PTMSP/PAN composite membranes were prepared by pouring PTMSP on PAN carrier, which had higher permeability than commercial nanofiltration membranes for ethanol, methanol and so on. Combined with the previous methods to improve the gas separation performance, Cheng et al. [56] significantly improved the permeability of ethanol and the rejection of rose red by adding PAF-1 and p-DCX. This series of studies have shown that PTMSP, a high free-volume glassy polymer, can be used not only for gas separation, but also for solvent separation.

In addition to PTMSP, some other substituted polyacetylenes were also used for solvent separation. In 1995, Masuda et al. [57] used [*o*-*w*(perfluorohexyl) phenylacetylene monomer to synthesize fluorinated polyphenylacetylene poly-(*o*-*n*-C_6_F_13_PA), which has ethanol pervaporation selectivity in ethanol and water pervaporation. Although the separation factor of ethanol and water is low, only 1.7, it can be seen from the chemical structures (see Figure 6) that most aromatic polyacetylenes are water pervaporation-selective, while fluorinated polyphenylacetylene exhibits ethanol pervaporation selectivity, which may be ascribed to the strong hydrophobicity of perfluorohexyl group. This also provides an idea to improve the pervaporation selectivity of alcohols by introducing strong hydrophobic groups. Additionally, they also synthesized silicon-containing poly [1-phenyl-2-(p-trimethylsilyl) phenylacetylene] and poly [1-β-naphthyl-2-(*p*-trimethylsilyl) phenylacetylene] [58], and desiliconized these two polymers. It was found that both polymers showed ethanol permeability selectivity before and after desiliconization. Usually, polyacetylene is soluble in low-polar benzene and n-hexane, so it is difficult to use polyacetylene membranes to separate benzene and n-hexane. In the pervaporation of benzene/cyclohexane, the desilication membrane has a low selectivity to benzene, but the permeation flux is large. This discovery also expands the application of substituted polyacetylene in the separation of low polar solvents such as benzene and n-hexane. In summary, based on the dissolution–diffusion mechanism, glassy polymers with nanopores can be used for gas and liquid separation.

## 5. Prospect

In 1983, Masuda reported gas and liquid separation capabilities of PTMSP, opened up a new direction for the applications of substituted polyacetylenes. In the subsequent research, focusing on gas separation and liquid separation, researchers synthesized a series of polyacetylene derivatives by changing substituents. Whether it is based on polyacetylenes or other materials, membrane separation is faced with the upper limit curve of selectivity and permeability and the problem of aging. Through the previous description and discussion, the selectivity can be improved by copolymerization, introduction of large substituents, crosslinking, halogenation, etc. For gas separation, without destroying the high free volume structure of the polyacetylene macromolecule, groups similar to the gas structure or capable of interacting with the gas can be introduced, and the selectivity to the gas can be improved by similar miscibility or interaction. For example, for the separation of carbon dioxide gas, ester groups can be introduced through a similar phase solubility mechanism, or amino groups can be introduced to interact with carbon dioxide to improve the solubility of carbon dioxide and other acidic gases, thus improving the selectivity. However, whether the polymerization reaction can be carried out after the introduction of groups is another difficult problem. It may be feasible to introduce polar groups by post-polymerization modification. In addition, for substituted polyacetylenes, the ratio of monomer to initiator, temperature, reaction time, etc., will affect the molecular weight and molecular weight distribution and then affect the mechanical strength of the separation membrane. When it comes to aging, porous additives can prevent the collapse of pore size and effectively solve the problem of physical aging on the basis of maintaining the high free volume of the polymer. From the more expensive PAF-1 to *p*-DCX, the cost reduction makes it further away from commercial applications. Similarly, for the membrane materials that substituted polyacetylenes, the current focus is mainly on improving its separation performance. If it is to be applied in practice, it is also necessary to consider how to reduce its cost. In summary, although a lot of research has been conducted on the gas and liquid separation of substituted polyacetylenes, there is still capacity for further research in membrane separation considering the above problems. Problems are always solved step by step. It is greatly hoped that in time, membrane materials based on substituted polyacetylenes can be put into use.

## Figures and Tables

**Figure 1 molecules-28-02748-f001:**
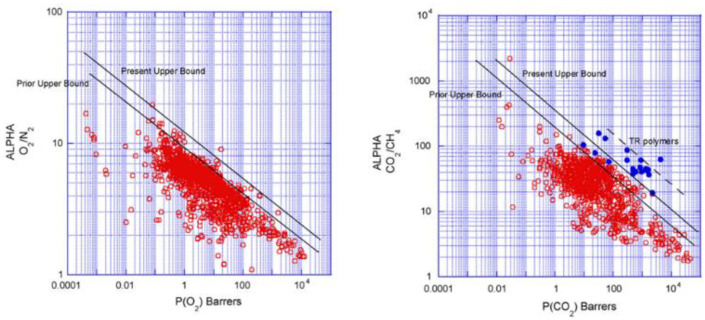
Upper bound correlation for O_2_/N_2_ separation, CO_2_/CH_4_ separation. Reproduced from Ref. [21] with permission from Elsevier, copyright 2008.

**Figure 2 molecules-28-02748-f002:**
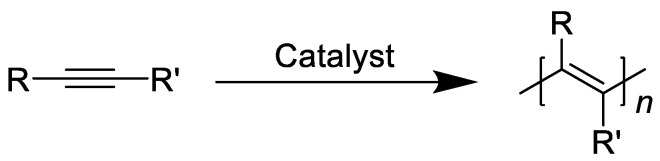
General synthetic route to substituted polyacetylenes.

**Figure 3 molecules-28-02748-f003:**
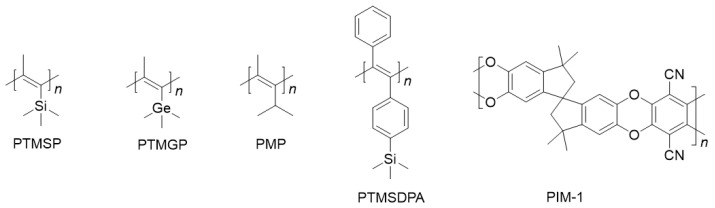
Representative disubstituted polyacetylenes and PIM-1 with good gas permeability.

**Figure 4 molecules-28-02748-f004:**
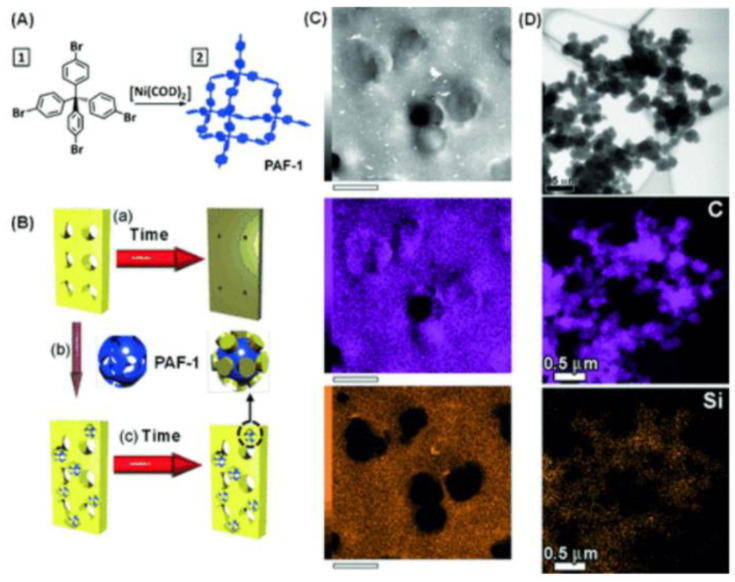
(**A**) Synthesis of PAF-1. (**B**) An illustration of super glassy polymer/PAF-1 intermixing. Typically (**a**) PTMSP, PMP and PIM-1 densify to give a non-permeable conformation, (**b**) but with the addition of PAF-1, (**c**) the original permeable structure is maintained. (**C**) TEM and EDX mapping (purple: carbon map; orange: silicon map) of 50 nm thin PTMSP/PAF-1 film show that PAF-1 particles (outlined by black circles in the Si map) are surrounded by Si, indicating the fine dispersion of PAF-1 within the PTMSP matrix. Scale bar is 300 nm. (**D**) TEM and energy-dispersive X-ray spectroscopy (EDX) analysis of pre-treated PAF-1 particles that were immersed in PTMSP solution show that Si is well-intercalated within the PAF-1 particles. Scale bar is 0.5 μm. These particles were washed thoroughly with chloroform prior to imaging. Reproduced from Ref. [44] with permission from Wiley, copyright 2014.

**Figure 5 molecules-28-02748-f005:**
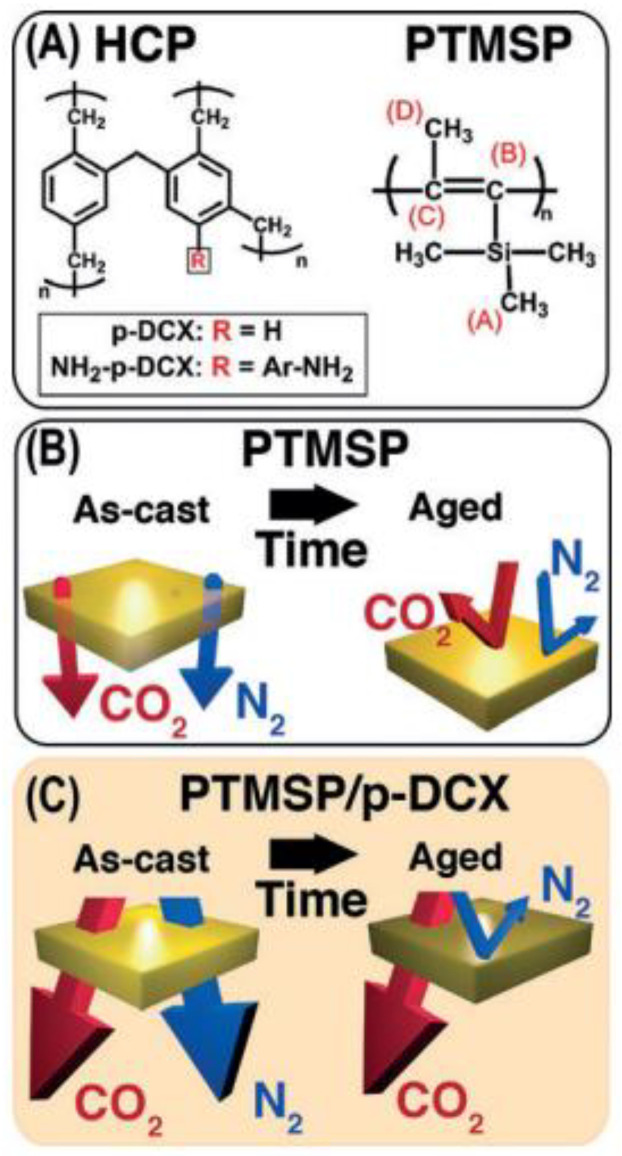
(**A**) Chemical structures of HCPs used in this work and PTMSP. (**B**) PTMSP membranes age over time and gas permeabilities are reduced. (**C**) By adding p-DCX into PTMSP gives a selective-aging membrane in which H_2_ and CO_2_ transport is preferred over CH_4_ and N_2_. Reproduced from Ref. [46] with permission from Wiley, copyright 2016.

**Figure 6 molecules-28-02748-f006:**
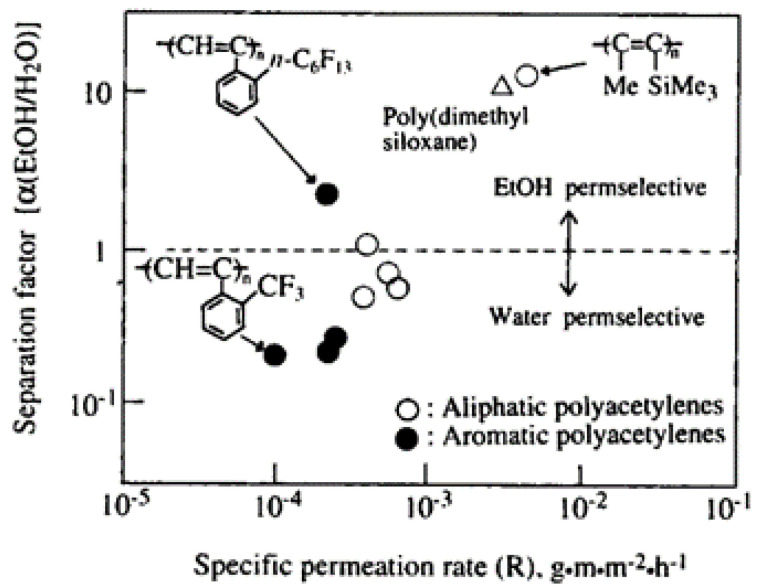
Plot of α(EtOH/H_2_O) vs. R of substituted polyacetylenes in pervaporation of EtOH/H_2_O mixture. Reproduced from Ref. [57] with permission from Wiley, copyright 1995.

## Data Availability

Not applicable.

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
