# Peer review of "Applications of Polyacetylene Derivatives in Gas and Liquid Separation"

_molecules, 2023, doi:10.3390/molecules28062748_

Round 1

Reviewer 1 Report

In this review, authors mainly introduced the application progress of substituted polyacetylene in gas and liquid separation and discussed the methods of enhancing separation selectivity and aging mechanism. The perspectives on the challenges and future development directions of this field were in-depth discussed. Specially, authors introduced many latest research progresses. It can help reader thoroughly know the past, now and future about the application of polyacetylenes. I suggest accept this manuscript for publication in Molecules after minor revisions. The suggestions and questions to the authors are as follows:

1.      Does the preparation method of substituted polyacetylene have an important to the performance of the separation membrane?

2.      More than once in the article, the introduction of similar groups or groups interacting with gases into substituted polyacetylene has been introduced. Suggest give an example.

Author Response

Reply: We appreciate your positive comments on our manuscript. The following are our answers to your question and suggestion.

  1. Does the preparation method of substituted polyacetylene have an importance to the performance of the separation membrane?

------Reply: Thank you very much for your instructive question. We think the preparation method of the polymers from substituted acetylene has certain influence on the separation performance of the membrane. But in fact, in the cited references, all of the samples were prepared by solution polymerization. The other commonly used polymerization methods such as emulsion polymerization, mass polymerization, and interface polymerization have not been applied to the polymerization of substituted acetylenes showing high efficiency of gas/liquid separation. Consequently, we have not described the effect of preparation method on the performance of the separation membrane.

  1. More than once in the article, the introduction of similar groups or groups interacting with gases into substituted polyacetylene has been introduced. Suggest give an example.

------Reply: Thank you very much for your good suggestion! For example, for the separation of carbon dioxide gas, ester groups can be introduced through a similar phase solubility mechanism, or amino groups can be introduced to interact with carbon dioxide to improve the solubility of carbon dioxide and other acidic gases, thus improving the selectivity. We have added these sentences into the perspective section of the revised manuscript. 

Reviewer 2 Report

This minor review introduced the development and application of substituted polyacetylene in gas separation and liquid separation based membrane technology. The paper was well organized and no more further review is required. 

A minor suggestion: adding the development of theoretical studies (like molecular simulation) on polyacetylene membranes  would be much better. 

Author Response

------Reply: Thank you for your constructive suggestion! We have added a short paragraph to describe the development of theoretical studies in this area. Please find the details as below and in the revised manuscript.

Molecular dynamics simulation can help researchers to predict and analyze the experimental results to a certain extent. For example, Catlow et al. [31] obtained potential parameters by using bond energy, vibration data and structural data to study the original form of trans-polyacetylene and its mobility under different doping. Molecular dynamics simulation can also be used to analyze the gas separation performance of substituted acetylene. Clough et al. [32] calculated the single-bond rotational potential barrier of several substituted acetylene by molecular dynamics simulation and calculated out that the chain conformation became non-planar with the increase of the volume and number of side groups. According to their results, the rotation potential barrier of TMSP around the single bond of PTMSP main chain is about 40 kcal/mol. Thus the torsional chain conformation will be maintained at room temperature. The dimension of the chain will change significantly only through the torsional wave in the narrow energy well. In a word, molecular dynamics simulation is an effective means to design a separation-capable substitute for polyacetylenes.